# F2: Let Large Language Models Think like Aristotle

## Abstract

With the wide application of Large Language Models (LLMs), the accuracy and reliability of the content they generate have become the focus of attention. The hallucination of the content generated by the large language model seriously affects the credibility and practicability of the model in key scenarios. However, the mainstream hallucination detection technology relies on external knowledge bases to verify the authenticity of the content generated by the model, or uses a large number of annotation data for training. These methods require complex model structure and support, will consume a lot of resources and time, and cross domain generalization ability is poor. In this paper, we proposes a new hallucination detection method, which allows the LLMs to imitate the way of thinking of the philosopher Aristotle. We decompose the complex hallucination verification process into two distinct subjects (called F2): a. **F**actual hallucination detection: verifying the fact finding of the content generated by the model and analyzing the optimal solution; b. **F**idelity hallucination detection: Logical verification based on classical logical forms, including the reasoning systems or logical forms of logic such as Aristotle's most outstanding contribution, syllogism. The experimental results show that this method not only improves the recognition and analysis ability of the LLMs itself for illusory content, but also enhances the interpretability of the defects of the LLMs, enabling the developers of the LLMs to effectively identify the sources of errors and improve the model capabilities.

## 1 Introduction

In recent years, with the expansion of model scale and the diversification of training data, LLMs have shown strong language understanding and generation ability (Brown et al., 2020). However, there is a hallucination in the generation process of LLMs, that is, the generated content seems reasonable in semantics, but is inconsistent with facts, context or external knowledge (Ji et al., 2023). For LLMs hallucinations, researchers have proposed a variety of detection and mitigation methods. Early work mainly relied on external knowledge base to detect hallucinations by comparing the generated content with the facts in the knowledge base (Guo et al., 2022). Recent studies have explored detection methods based on the internal mechanism of the model, such as using attention weight, intermediate representation or uncertainty estimation to identify potential hallucinations (Farquhar et al., 2024). Recently, some researches have divided the complex reasoning process into memory and reasoning, and used the learnable control token to solve the task (Jin et al., 2024).

This paper proposes a novel hallucination detection framework that enables large language models (LLMs) to emulate the cognitive process of philosophers exemplified by Aristotle in their pursuit of truth and logical reasoning to investigate the nature of reality and human affairs. The framework decomposes the complex hallucination verification process into two equally important and distinct components: factual hallucination detection and fidelity hallucination detection. Factual hallucination detection corresponds to the exploration of truth, which involves examining and defining entities and their attributes from multiple perspectives to verify the objective essence of people and objects. Fidelity hallucination detection corresponds to logical reasoning, focusing primarily on the relationships between people and things to validate the external logic governing these connections. This method not only innovatively and systematically defines the focus in the field of hallucination detection, but also enhances the interpretability of the capability deficiencies of LLMs, providing

constructive suggestions for developers of LLMs to effectively identify the sources of errors and improve the model's capabilities.

The experimental results show that this method can improve the recognition and analysis ability of the hallucination content of the model, and enhance the reasoning ability of the model. Our method achieves 89.52% and 92.66% accuracy on CommonsenseQA (Talmor et al., 2019) and QASC dataset (Khot et al., 2019), respectively, using Qwen3-8B (Yang et al., 2025), representing remarkable improvements of 12.62% and 12.53% over the zero-shot baseline. On StrategyQA (Geva et al., 2021), it achieves 76.35% accuracy with Gemma3-12B (Team et al., 2025). On HaluEval (Li et al., 2023), our method obtains an F1 score of 0.894 using Qwen3-8B, surpassing the zero-shot approach by 0.18. More notably, our method consistently outperforms the results of CoT (Wei et al., 2022), CoT-SC (Wang et al., 2023), and ToT (Yao et al., 2023) under the same experimental settings across all four datasets.

Our main contributions are as follows:

- A new approach to hallucination detection: inspired by Aristotle and other philosophers, we innovatively integrate factual exploration and logical inference and applied them to hallucination detection field.

- New self-verification framework for hallucinations of LLMs: We propose a resource-saving, transferable and adaptive hallucination self verification method for LLMs without training.

- Improving benchmark performance of LLMs: the proposed framework achieved competitive improvements in accuracy on StrategyQA, CommonsenseQA, QASC and HaluEval dataset.

## 2 RELATED WORK

### 2.1 HALLUCINATION DETECTION

Early hallucination detection in neural models often relied on symbolic reasoning layers that sit on top of a neural encoder. DeepProbLog (Manhaeve et al., 2018) and NeurASP (Yang et al., 2023), for example, treat the neural output as a distribution over discrete symbols and feed it to a symbolic solver, a strategy that yields faithful reasoning but incurs heavy computational overhead. Patrick (Lewis et al., 2020) explored a general fine-tuning method of Retrieval-Augmented Generation (RAG) by combining the language generation model of pretrained parametric and non parametric memory.

More recent work has shifted toward scalable, fully-neural strategies. SelfCheckGPT (Manakul et al., 2023) detects factual hallucinations by asking the same LLM to sample multiple continuations and then measuring internal consistency via entailment scoring, achieving zero-resource black-box detection. Huang et al. (Huang et al., 2025) propose a taxonomy that distinguishes intrinsic vs. extrinsic hallucinations and benchmark existing detectors across open-ended generation tasks. Chen et al. (Chen et al., 2024) extend detection to the multimodal regime, introducing unified frameworks that jointly assess object, attribute, and relation-level hallucinations in large vision–language models.

### 2.2 INFERENCE IN LARGE LANGUAGE MODELS

Early work on inference in Large Language Models (LLMs) focused on exact or variational methods for small transformer decoders (Vaswani et al., 2017). Exact posterior inference over exponentially-large output spaces is infeasible, so beam search (Graves, 2012) and temperature-controlled sampling (Holtzman et al., 2020) became the de facto decoding strategies.

In recent research, a complementary line of research treats inference as amortised optimisation. In-context learning (Brown et al., 2020) reframes few-shot tasks as forward passes that implicitly perform gradient-descent in the model's activation space (Holtzman et al., 2020). The scaling law of test time (Snell et al., 2024) shows that increasing the calculation budget of reasoning time through CoT sampling, verifier reordering or verifiable reward model can generate predictable performance gains in reasoning benchmark.

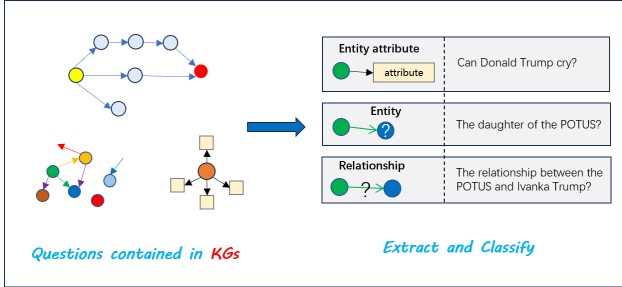

Figure 1: Abstraction of real world questions and classification. The answers to these questions are contained in a growing knowledge graph cluster, but the classification of these questions can be divided into attribute, entity and relationship according to the target subject.

In addition, the CoT (Wei et al., 2022) suggests that piecewise linear nodes are introduced between the input and output, and the reasoning process of the input and output is connected in the form of a chain. The later improved CoT-SC (Wang et al., 2023) generates multiple thought chains, and then takes most of the answers as the final answer. CoT derived Tree-of-Thoughts (ToT) (Yao et al., 2023) builds a tree structure to break the independence between CoT-SC multi chains, and gives the model the ability to search among multiple reasoning chains through the tree structure. Graph of Thoughts (GoT) (Besta et al., 2024) generalizes the structure of the tree. Any two reasoning nodes may be connected by reasoning chains, so each reasoning node can have multiple parent nodes and child nodes. By aggregating these nodes, the sub graph for solving the sub problem is obtained, and the final solution of the problem is formed after the combination.

## 3 METHODOLOGY

### 3.1 BACKGROUND SETTING

In real-world application scenarios, the answer to a problem is actually contained in a library $KG\{g_1, g_2, \ldots, g_m\}$ with an infinite number of large knowledge graphs, where each node of the knowledge graph represents an entity and attribute, and each edge represents a relationship. Assuming that the user wants to ask a question from a set $Query\{q_1, q_2, \ldots, q_n\}$, each element of the set represents a question, and the elements in this set, which are these questions, can be roughly abstracted into three types as shown in Fig. 1: questions about attributes, questions about entities, and questions about relationships. At the same time, the process of the big language model inferring the answers $Answer\{a_1, a_2, \ldots, a_n\}$ to the problem set $Query\{q_1, q_2, \ldots, q_n\}$ is actually the process of generating a knowledge graph $KG'\{g_1', g_2', \ldots, g_n'\}$. The process of verifying the answers $Answer\{a_1, a_2, \ldots, a_n\}$ is actually the process of verifying whether the element g' in the generated knowledge graph $KG'\{g_1', g_2', \ldots, g_n'\}$ is an element in the knowledge graph $KG\{g_1, g_2, \ldots, g_m\}$. Since the elements in KG and KG' are knowledge graphs, it is necessary not only to verify the nodes and edges of the graph, but also to verify the path from the original node to the final node.

The above considerations can ultimately be summarized as reflections on and a conceptualization of the nature of reality, aligning with the fundamental question of philosophy, the relationship between matter and consciousness. These precisely correspond to the purposes of factual hallucination detection and fidelity hallucination detection, respectively.

### 3.2 FACTUAL HALLUCINATION DETECTION

Factual hallucination detection is an analysis of internal factors of things, mainly the various attributes of the subject entity, including the state, characteristics, properties, feelings, etc. of people and things. The main process of factual hallucination detection is divided into two parts: verification of factual correctness and analysis of optimal solutions. The first step is the simplest verification of factual correctness, which requires LLM to make direct judgments on entities and their attributes based on its internal knowledge base (language patterns and factual knowledge learned during train-

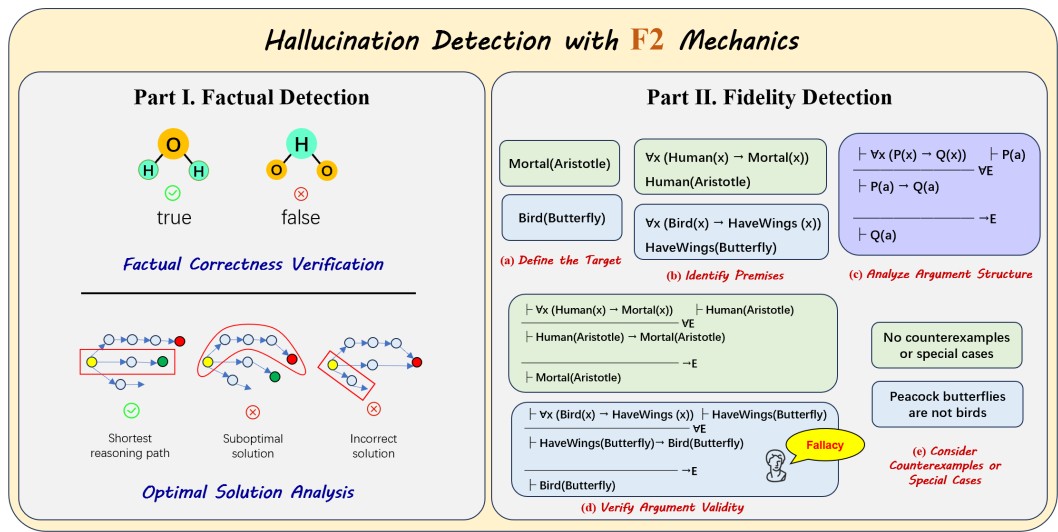

Figure 2: Frame design of F2 mechanics. It consists of two parts: Factual Hallucination Detection part: factual correctness analysis and optimal solution analysis; Fidelity Hallucination Detection part: validate the logic inference, using "Aristotle is mortal" as a positive sample and "Butterfly is a bird" as a negative sample for illustration.

ing). We represent this part as:

$$\mathcal{F} = \mathcal{F}_{\text{FactualCorrectness}}(x, \text{reason}), \quad \mathcal{F} \in \{\text{true}, \text{false}\}, \tag{1}$$

where $\mathcal{F}$ represents the result of judging the correctness of facts. For the example part in Fig. 2, the internal knowledge base of LLM contains the molecular structure of water, which is $H_2O$. However, due to deviations in the Transformer's capture of semantic information and contextual relationships in the input problem, LLM finally outputs the structure of $O_2H$. At this point, simple retrieval and analysis are needed to detect this hallucination problem that does not conform to general facts and objective laws.

The Optimal Solution Analysis section is specifically designed for a particular scenario: the answer to the question is open-ended, but the goal is to find the one among the possible answers that best aligns with the meaning conveyed by the question. We denote this section as:

$$\mathcal{O} = \mathcal{O}_{\text{OptimalSolution}}(y, \text{reason}), \quad \mathcal{O} \in \{\text{true}, \text{false}\}, \tag{2}$$

where $\mathcal{O}$ denotes the judgment result of optimal solution analysis. For example, when someone asks what can extinguish a fire, A says water, B says coffee, and C says cola. Among these options, the most suitable option is A, as water is a very common and effective fire extinguishing agent. However, strictly speaking, the other options are not incorrect, just less suitable than water. As shown in Fig. 2, when there are multiple valid explanations, the optimal solution analysis part needs to verify whether LLM's answer is the simplest, most direct, and objective answer (i.e. the answer with the shortest inference path). The results of factual hallucination detection are expressed as:

$$f_{v1} = \begin{cases} \text{true}, & \text{if } \mathcal{F} = \text{true and } \mathcal{O} = \text{true}, \\ \text{false}, & \text{otherwise}, \end{cases} \tag{3}$$

where $f_{v1}$ represents the verification result of factual hallucination detection.

### 3.3 FIDELITY HALLUCINATION DETECTION

Philosophical inquiry is premised on the use of logical thought and deduction to examine fundamental questions. This approach is imperative due to the constraints of human understanding, which

often necessitate drawing conclusions via inference from available information when direct solutions are unattainable. In a parallel manner, there are instances where an LLM cannot locate specific entities, their properties, or inter-entity relationships within its knowledge repository, thus requiring it to reason through parental or analogous entities. This parallel underpins the adoption of logical verification, the results of which can, in turn, corroborate the accuracy of factual validation.

The fidelity hallucination detection is an analysis of the connections between things, including actions, relationships, etc. At this stage, our main focus is on verifying the process by which LLM infers answers, that is, validating the reasoning logic.

This part of verification is divided into five steps, as shown on the right side of Fig. 2, using "Aristotle is mortal" as a positive sample for illustration:

Step 1: Define the Target. This step identifies the conclusion "Aristotle is mortal" that needs to be proved and expressed as the atomic formula $Mortal(Aristotle)$ of predicate logic.

Step 2: Identify Premises. In this step, the major premise "All humans are mortal" and the minor premise "Aristotle is a human" have been identified and expressed in formal logic as $\forall x \, (\text{Human}(x) \rightarrow \text{Mortal}(x))$ and $Human(Aristotle)$, respectively.

Step 3: Analyze Argument Structure. Determine the method of valid deductive reasoning proof using natural deduction in predicate logic, that is, the inference path (The purple part in the figure).

Step 4: Verify Argument Validity. Substitute the premises obtained in Step 2 into the inference path derived in Step 3 and verify. Both premises are valid, so the conclusion "Aristotle is mortal" is successfully derived.

Step 5: Consider Counterexamples or Special Cases. In this positive sample, there is no counterexample that can refute the conclusion.

Similarly, for the negative sample "Butterfly is a bird", the first three steps are the same as in the positive sample. However, in the verification process of Step 4, a typical logical fallacy "affirming the consequent" is demonstrated, which ultimately leads to a FAIL result in the faithfulness hallucination detection. Additionally, Step 5 produces a counterexample "Peacock butterflies are not birds" that also refutes the reasoning outcome from Step 4.

These five steps can be expressed as follows:

$$\mathcal{P} = \mathcal{P}_{\text{premise}}(P(x), Q(y), R(z), \ldots), \quad \mathcal{P} \in \{\text{true}, \text{false}\}, \tag{4}$$

$$\mathcal{I} = \mathcal{I}_{\text{inference}}(P(a) \rightarrow Q(a), \ldots), \quad \mathcal{I} \in \{\text{valid}, \text{invalid}\}, \tag{5}$$

$$\mathcal{E} = \mathcal{E}_{\text{counterexample}}(\varepsilon), \tag{6}$$

where $\mathcal{P}$ denotes the truth of all premises, $\mathcal{I}$ denotes the validity of the inference process, and $\mathcal{E}$ denotes the existence of counterexamples; if no counterexample exists, $\mathcal{E}$ is empty.

Based on these processes, we obtained the result of fidelity hallucination detection:

$$f_{v2} = \begin{cases} \text{true,} & \begin{aligned} &\text{if } \mathcal{P} = \text{true} \\ &\text{and } \mathcal{I} = \text{valid} \\ &\text{and } \mathcal{E} = \text{NULL,} \end{aligned} \\ \text{false,} & \text{otherwise,} \end{cases} \tag{7}$$

where $f_{v2}$ represents the verification result of fidelity hallucination detection.

### 3.4 Retry Mechanism

After obtaining the results of factual hallucination detection ($f_{v1}$) and fidelity hallucination detection ($f_{v2}$), if at least one of them is not true, we will use the result of verification failure and a concise reason as prompt words to retry the original problem. After receiving a new response, perform a complete factual hallucination detection and fidelity hallucination detection again.

# 4 EXPERIMENTS

## 4.1 EXPERIMENT SETUP

### 4.1.1 DATASETS.

Our experiments was conducted on three datasets: CommonsenseQA (Talmor et al., 2019) is a new multiple-choice question answering dataset that requires different types of commonsense knowledge to predict the correct answers . It contains 12,102 questions with one correct answer and four distractor answers. The StrategyQA (Geva et al., 2021) dataset was created through a crowdsourcing pipeline for eliciting creative and diverse yes/no questions that require implicit reasoning steps. To solve questions in StrategyQA, the reasoning steps should be inferred using a strategy. QASC (Khot et al., 2019) is a question-answering dataset with a focus on sentence composition. It consists of 9,980 8-way multiple-choice questions about grade school science (8,134 train, 926 dev, 920 test), and comes with a corpus of 17M sentences. HaluEval (Li et al., 2023) includes 5,000 general user queries with ChatGPT responses and 30,000 task-specific examples from three tasks, i.e., question answering, knowledge-grounded dialogue, and text summarization.

### 4.1.2 MODELS.

To comprehensively evaluate the effectiveness of our method on the selected dataset, we used several highly accepted and latest open-source models: Mistral-7B-Instruct-v0.3 (Jiang et al., 2023), Llama-3.1-8B-Instruct (Dubey et al., 2024), Qwen3-8B (Yang et al., 2025), Gemma-3-12b-it (Team et al., 2025).

### 4.1.3 BASELINES.

Since our method is based on the performance of the LLM itself, we use zero-shot learning (Xian et al., 2017), CoT prompts (Wei et al., 2022), CoT-SC prompts (Wang et al., 2023) and ToT prompts (Yao et al., 2023) as the baseline for the experiments.

### 4.1.4 EVALUATION METRIC.

We use accuracy to evaluate the performance of all methods on three datasets except HaluEval dataset. For HaluEval dataset, we use F1 Score for evaluation.Details of the extraction and verification of the answers are provided in the appendix.

## 4.2 MAIN RESULTS

Compare our method with the following baseline methods:zero-shot learning, CoT, CoT-SC and ToT. In Tab. 1 and Fig. 3, we demonstrate the accuracy or F1 score of these methods, and our approach achieved significant improvements on all datasets and models. It is worth noting that our method achieved the highest scores on Qwen3-8B for the CommonsenseQA, QASC and HaluEval dataset, with scores of 89.52%, 92.66% and 0.894, respectively. Compared to the results of zero shot learning (76.9%, 80.13% and 0.714), we achieved improvements of 12.62%, 12.53% and 0.18, respectively. And our method is better than most baseline methods in the same configuration. These results demonstrate the strong competitiveness of our method in the fields of hallucination detection and improving model performance.

However, for the CommonsenseQA dataset and QASC dataset, the performance of CoT, CoT-SC and ToT methods on Qwen3-8B actually decreased compared to direct response. This may be due to the fact that the CommonsenseQA dataset and QASC dataset are multiple-choice question answering datasets, while these methods are based on chain inference, terminating inference after obtaining possible answers and not focusing on the optimal choice, which further demonstrates the superiority of our method.

For the StrategyQA dataset, our method achieves the best performance with Gemma3-12B (76.35%). However, due to the fact that the StrategyQA dataset is based on binary judgments of Wikipedia's problems and relies more on the textual data encountered during the training process, a one-step inference cannot form an inference chain, which is not rigorous on the evidence chain and cannot

Table 1: Main Comparative Experiment Results. The comparative experiments were divided into five groups: Zero-shot, CoT, CoT-SC, ToT, and our method (F2). The best results are marked in **bold**, and the second-best are underlined.

| Methods | Models | StrategyQA | CommonsenseQA | QASC | HaluEval | Average |
|---|---|---|---|---|---|---|
| Zero-shot | Mistral-7B | 0.5800 | 0.730 | 0.6609 | 0.678 | 0.6622 |
| | Llama3.1-8B | 0.5968 | 0.747 | 0.7927 | 0.752 | 0.7221 |
| | Qwen3-8B | 0.6390 | 0.769 | 0.8013 | 0.714 | 0.7308 |
| | Gemma3-12B | 0.7263 | 0.768 | 0.7991 | 0.782 | 0.7688 |
| CoT | Mistral-7B | 0.5727 | 0.713 | 0.6706 | 0.683 | 0.6598 |
| | Llama3.1-8B | 0.5284 | 0.714 | 0.7981 | 0.752 | 0.6981 |
| | Qwen3-8B | 0.7229 | 0.647 | 0.6458 | 0.794 | 0.7024 |
| | Gemma3-12B | 0.7562 | 0.781 | 0.8261 | 0.755 | 0.7795 |
| CoT-SC | Mistral-7B | 0.5718 | 0.711 | 0.6782 | 0.693 | 0.6635 |
| | Llama3.1-8B | 0.5415 | 0.747 | 0.8326 | 0.737 | 0.7145 |
| | Qwen3-8B | 0.7265 | 0.667 | 0.6598 | 0.800 | 0.7133 |
| | Gemma3-12B | 0.7569 | 0.776 | 0.8272 | 0.761 | 0.7802 |
| ToT | Mistral-7B | 0.6326 | 0.483 | 0.4600 | 0.678 | 0.5634 |
| | Llama3.1-8B | 0.5904 | 0.700 | 0.6582 | 0.730 | 0.6696 |
| | Qwen3-8B | 0.7036 | 0.755 | 0.7883 | 0.833 | 0.7700 |
| | Gemma3-12B | 0.7342 | 0.748 | 0.7927 | 0.782 | 0.7667 |
| Ours (F2) | Mistral-7B | 0.6071 | 0.7535 | 0.7181 | 0.723 | 0.7004 |
| | Llama3.1-8B | 0.6571 | 0.7867 | 0.8200 | 0.781 | 0.7542 |
| | Qwen3-8B | 0.7333 | **0.8952** | **0.9266** | **0.894** | **0.8623** |
| | Gemma3-12B | **0.7635** | 0.8029 | 0.8817 | 0.845 | 0.8232 |

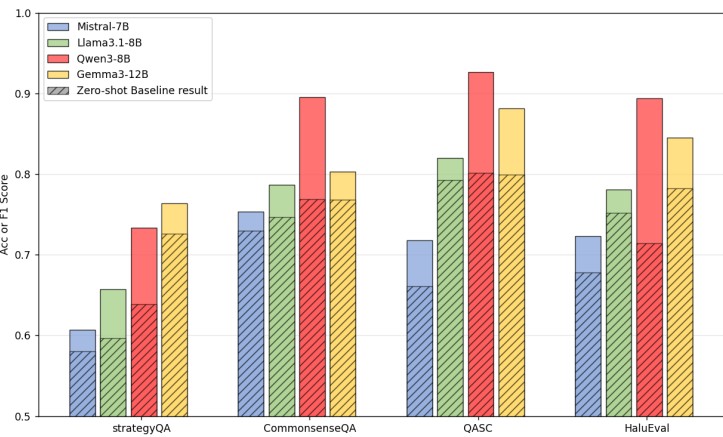

Figure 3: Performance comparisons of F2 utilizing different LLMs on three datasets.

fully utilize the advantages of our method. The lower accuracy of all methods on the StrategyQA dataset compared to other datasets also proves this point.

## 4.3 ABLATION STUDY

In the ablation experiment stage, we separately blocked the effects of factual hallucination detection and fidelity hallucination detection, and studied the impact and effectiveness of these two parts.

Table 2: Ablation Study on Llama3.1-8B and Qwen3-8B Across Three Benchmarks.

| | StrategyQA | CommonsenseQA | QASC | Average |
|---|---|---|---|---|
| **Llama3.1-8B** | | | | |
| F2 | **0.6571** | **0.7867** | **0.8200** | **0.7546** |
| Factual_only | 0.6244 | 0.7451 | 0.8034 | 0.7243 |
| Fidelity_only | 0.6369 | 0.7447 | 0.8023 | 0.7280 |
| **Qwen3-8B** | | | | |
| F2 | **0.7333** | **0.8952** | **0.9266** | **0.8517** |
| Factual_only | 0.6343 | 0.8264 | 0.8398 | 0.7668 |
| Fidelity_only | 0.7059 | 0.8349 | 0.9030 | 0.8146 |

### 4.3.1 IMPACT OF FIDELITY HALLUCINATION DETECTION.

In this section, we will mask fidelity hallucination detection and only use factual hallucination detection to observe the impact of fidelity hallucination detection on model performance. We compared the performance of two models with significant performance differences (LLaMA-3.1-8B and Qwen3-8B) on three datasets (StrategyQA, CommonsenseQA and QASC), and the results are shown in the factual_only section of Tab. 2. Overall, the accuracy has decreased as expected compared to our method, especially with Qwen3-8B showing a 9.9% (StrategyQA), 6.88% (CommonsenseQA) and 8.68% (QASC) decrease in performance on the three datasets, respectively.

### 4.3.2 IMPACT OF FACTUAL HALLUCINATION DETECTION.

In this section, we will mask factual hallucination detection and only use fidelity hallucination detection to observe the impact of factual hallucination detection on model performance. Similar to the study on the impact of fidelity hallucination detection in 4.3.1, we compared the performance of two models (LLaMA-3.1-8B and Qwen3-8B) with significant performance differences on three datasets (StrategyQA, CommonsenseQA and QASC). The results obtained are shown in the fidelity only section of Table 2. Overall, the accuracy has decreased as expected compared to our method.

### 4.3.3 COMPREHENSIVE IMPACT.

In fact, by analyzing the data in Tab. 1 and 2, a very interesting phenomenon can be found. Regarding the performance of LLaMA-3.1-8B on the CommonsenseQA and QASC datasets, the difference between the results of factual hallucination detection alone or fidelity hallucination detection alone and zero shot is very small. However, under the combined effect of factual hallucination detection and fidelity hallucination detection, the accuracy of LLaMA-3.1-8B on the CommonsenseQA dataset has been improved. In addition, for these three datasets, the difference in performance between LLaMA-3.1-8B using either individual factual hallucination detection or individual fidelity hallucination detection is also very small. However, similarly, under the combined effect of factual hallucination detection and fidelity hallucination detection, the performance has been improved on both datasets. Meanwhile, as shown in Tab. 2, for the higher performance Qwen3-8B, the effect of fidelity hallucination detection is more advantageous than factual hallucination detection.

### 4.4 CASE STUDY

Fig. 4 shows the case study in the StrategyQA dataset. First, we performed a factual hallucination detection. By analyzing the contents in the figure, we can find that LLM mistakenly identified Petroleum (CAS 8009-03-8) as sodium lauryl sulfate (SLS)(CAS 151-21-3). For people whose skin barrier has been damaged, SLS may further stimulate the skin. So LLM answered true. But in fact, CAS 8009-03-8 (Petrolatum) is usually not harmful to rash, and can even be used to relieve rash. Therefore, the verification result of the factual hallucination test is FAIL.

For fidelity hallucination detection, we carry out logical verification according to the five steps shown in the figure. Step 1 is to let LLM determine the goal, that is, we need to prove the cor-

> **Question:** Is CAS number 8009-03-8 harmful for a rash?
> "Answer": "true"
> Ground truth: "false"

> **Factual verification:**
> "verified": FAIL,
> "reason": First, I should figure out what CAS number 8009-03-8 refers to. CAS numbers are unique identifiers for chemical substances. From what I recall, CAS 8009-03-8 is for Sodium Lauryl Sulfate (SLS). Let me consider that maybe the answer is incorrect because the CAS number provided doesn't correspond to a known harmful substance for rashes. Wait, I think the correct CAS number for Sodium Lauryl Sulfate is indeed different. CAS 8009-03-8 is the registry number for mineral oil (heavy) or paraffinum liquidum, a petroleum-derived mixture often used in shampoos, soaps, and toothpaste. It is generally considered safe for topical use when highly refined.

> **Fidelity verification:**
> "verified": PASS,
> "reason":
> Step 1: Define the Target: The question is whether the CAS number 8009-03-8 is harmful for a rash, and the predicted answer is TRUE.
> Step 2: Identify Premises and Conclusion: CAS 8009-03-8 is for Sodium Lauryl Sulfate (SLS). So the argument would be: SLS is an irritant (premise), irritants can cause rashes (premise), therefore SLS can cause a rash (conclusion), hence harmful for a rash (TRUE).
> Step 3: Analyze Argument Structure: Assuming that the premises are that SLS is a known skin irritant and that irritants can cause rashes, then the conclusion that SLS is harmful for a rash (i.e., can cause a rash) would logically follow.
> Step 4: Verify Argument Validity: This is a classic syllogism. The structure is valid.
> Step 5: Consider Counterexamples or Special Cases: SLS is a common surfactant used in many personal care products like shampoos, soaps, and toothpaste.

Figure 4: Case study on the StrategyQA dataset using Qwen3-8B. For this question, the LLM's response is inconsistent with the ground truth, leading to a FAIL in factual verification. However, the response receives a PASS in fidelity verification.

rectness of the argument that CAS 8009-03-8 is harmful to rash. Step 2 is to analyze the premise and conclusion. In this step, LLM first replaced CAS 8009-03-8 with Solid Lauryl Sulfate (SLS) equivalently. Although this is not correct in fact, it is not a problem in logic. Then LLM marks two premises and a conclusion. Step 3 is to analyze and demonstrate the structure. LLM selects Aristotle's classical syllogism for logical verification. The major premise is that the stimulant can cause skin rash, and the minor premise is that SLS is a stimulant. The conclusion is that SLS can cause skin rash and is harmful to skin rash. Step 4 is to verify the validity of the parameters. It is obvious that this is a classic syllogism structure. This argument is valid. Therefore, the logical verification will pass. Step 5: consider counterexamples and special cases. Here LLM considers the application of personal care products, but the context of the topic itself is a special case. In general, SLS is widely used and usually safe, but for some people, especially those with sensitive skin or existing skin problems, it may aggravate the rash. So in this step, this general situation can not affect the final result, so in general, the verification result of fidelity illusion detection is PASS.

## 5 LIMITATION

Our large language model hallucination detection method has shown good results and competitive advantages in the field of hallucination detection, but there are still some limitations. First of all, for specific problems that only need factual hallucination detection or only need faithful hallucination detection, we cannot judge dynamically, so we are forced to carry out two kinds of verification at the same time. This is also proved in the appendix.

## 6 CONCLUSION

This paper proposes a new hallucination detection mechanics, which allows LLM to imitate the way of thinking of philosopher Aristotle and decompose the complex hallucination detection process into two clear subjects. The factual hallucination detection part mainly verifies whether the original response of the model conforms to the general facts and objective laws it has learned (verifying correctness of the answer), and determines whether the answer given by the model is the most appropriate choice with the shortest reasoning path among all effective explanations. The fidelity hallucination detection part verifies whether the answer of the model conforms to the reasoning logic (verifying the reasoning process). This hallucination detection method is training-free, offers significant advantages in terms of portability and adaptability and enhances the interpretability of the defects of the model.

## REPRODUCIBILITY STATEMENT

We promise to ensure the reproducibility of experimental results, disclose all experimental codes, specify the data processing process and dependent environment, and provide clear operation guidelines; All experiments are based on public data sets, and all experimental details that need attention are fully described in the text or appendix to ensure that others can reproduce the results of this paper under the same hardware conditions.

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

APPENDIX

## A  EXPERIMENT DETAILS

### A.1  TRAINING CONFIGURATION

All experiments involving Gemma-12B were conducted on a single NVIDIA A100 Tensor Core GPU with 80 GB HBM2. Experiments involving the Mistral-7B-Instruct-v0.3, Llama-3.1-8B-Instruct, and Qwen3-8B models were all performed on a single NVIDIA A40 GPU. All code was written in Python.

### A.2  PARAMETER SETTING

The sampling parameters are set as follows:

Listing 1: Generation Hyperparameters

```
temperature = 0.7
max_tokens = 512
top_p = 0.9
stop = None
```

### A.3  GENERATION OF DATA TO BE VERIFIED

We directly input the question to the LLM, and extract the specific answer of the model after obtaining the raw response. For example, in the CommonsenseQA and QASC datasets, we extract the options, and in the StrategyQA dataset, we extract the judgment results (true and false). After that, we substitute the extracted model answers into the questions, and submit the combined questions to our hallucination detection mechanism for verification.

### A.4  DATA EXTRACTION AND VERIFICATION

The LLM is guided, through the use of designed regular expressions and prompt templates with special tokens, to output the model-generated answer in an ideal format for comparison with a standard answer. This formatted output is subsequently extracted as the model_answer field, while the original response is preserved as raw_answer or answer_content. Evaluation is ultimately performed by computing accuracy or F1 scores based on a comparison between the model_answer and the standard answer.

## B  PROMPT TEMPLATE

### B.1  FACTUAL HALLUCINATION DETECTION

The prompt template of factual hallucination detection is shown in Fig. 5. We divide the factual hallucination detection into two parts, the verification of the correctness of facts and the analysis of the optimal solution.

### B.2  FIDELITY HALLUCINATION DETECTION

The prompt template of factual hallucination detection is shown in Fig. 6, which is verified in five steps.

## C  SUPPLEMENT RESULTS

### C.1  INFLUENCE OF FEW-SHOT

On the basis of the main experiment, we studied the impact of feed shot on our method, and adjusted the original results of zero-shot verification with hallucination detection to the results of feed shot

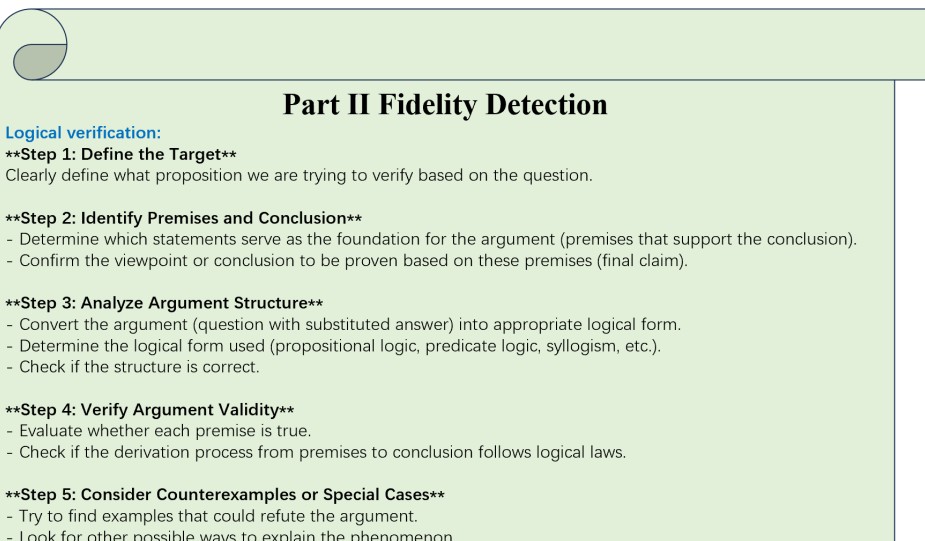

Figure 5: prompt template for factual hallucination detection.

Figure 6: prompt template for fidelity hallucination detection.

verification with hallucination detection. The comparison of experimental results is shown in Tab. 3.

The experimental results show that feed shot has little effect on our method, and the improvement of the accuracy of the final results on llama3.1-8B is limited, and even has a negative effect on mistral-7B and qwen3-8B.

## C.2 TEST OF FIDELITY VERIFICATION ON MODEL PERFORMANCE

We have tried to set the factual hallucination detection as necessary and the faithful hallucination detection as optional, and let LLM analyze and decide whether to carry out faithful hallucination detection. We carried out this experiment with llama3.1-8B on the CommonsenseQA dataset. The experimental results show that after making llama3.1-8B decide whether to perform fidelity hallucination detection, the accuracy obtained on the commonsenseqa data set is 73.96%, which is reduced

Table 3: Study on the Effect of Few-Shot Examples in Our Method. Experiments conducted on two benchmarks using three models of similar size. Best results in **bold**, second-best underlined.

| Methods | Models | StrategyQA | CommonsenseQA | QASC | Average |
|---|---|---|---|---|---|
| Zero-shot | Mistral-7B | 0.5800 | 0.730 | 0.6609 | 0.6569 |
| | Llama3.1-8B | 0.5968 | 0.747 | 0.7927 | 0.7121 |
| | Qwen3-8B | 0.6390 | 0.769 | 0.8013 | 0.7364 |
| | Gemma3-12B | 0.7263 | 0.768 | 0.7991 | 0.7644 |
| Few-shot | Mistral-7B | 0.6310 | 0.722 | 0.6901 | 0.6810 |
| | Llama3.1-8B | 0.6419 | 0.762 | 0.7927 | 0.7322 |
| | Qwen3-8B | 0.6450 | 0.798 | 0.8164 | 0.7531 |
| | Gemma3-12B | 0.7031 | 0.774 | 0.8164 | 0.7645 |
| Zero-shot + F2 | Mistral-7B | 0.6071 | 0.7535 | 0.7181 | 0.6929 |
| | Llama3.1-8B | 0.6571 | 0.7867 | 0.8200 | 0.7452 |
| | Qwen3-8B | 0.7333 | **0.8952** | **0.9266** | **0.8517** |
| | Gemma3-12B | **0.7635** | 0.8029 | 0.8817 | 0.8160 |
| Few-shot + F2 | Mistral-7B | 0.5900 | 0.7521 | - | - |
| | Llama3.1-8B | 0.6599 | 0.8019 | - | - |
| | Qwen3-8B | 0.7335 | 0.8895 | - | - |
| | Gemma3-12B | - | - | - | - |

Table 4: Ablation Study on Fidelity Verification for Llama3.1-8B on CommonsenseQA. Factual hallucination detection is required; fidelity (faithful) hallucination detection is varied as ablated component. Best result in **bold**.

| Method | CommonsenseQA |
|---|---|
| **Llama3.1-8B** | |
| F2 | **0.7867** |
| Factual_only | 0.7451 |
| Fidelity_only | 0.7447 |
| Fidelity_optional | 0.7396 |

to varying degrees compared with our method (F2) and the results of only performing factual hallucination detection and fidelity hallucination detection, as shown in Tab. 4.

# D    CASE STUDY OF QASC DATASET

It is worth noting that all the problems in the strategyqa dataset are binary judgment problems, which cannot play a real role in the optimal solution verification on this dataset, so we have omitted this part from this dataset. As for the analysis of optimal solution and the case study of other data sets, we will show it in the appendix in combination with other data sets.

Fig. 7 shows a case study of the QASC dataset. First, we performed a factual hallucination detection. By analyzing the contents in the figure, we can get that for the correctness of facts, climate (option B) refers to the long-term average of weather conditions, including temperature and humidity (which relates to water in the air). This aligns with general facts.

For optimal solution analysis, other options like storms (option A) are short-term weather events, mass (option C) and density (option F) are physical properties, seasonal (option D) and winter (option E) are time-based categories, length (option G) and fluid (option H) are unrelated. Climate is the most appropriate and objective answer as it directly encompasses temperature and atmospheric water content over time.

Question: What is described in terms of temperature and water in the air?
Choices: A. storms, B. climate, C. mass, D. seasonal, E. winter, F. density, G. length, H. fluid
Model Answer: B. climate
Ground truth: B. climate

**Factual verification:**
"verified": PASS,
"reason":
**Factual Correctness:** The question asks about what is described in terms of temperature and water in the air. Climate (B) refers to the long-term average of weather conditions, including temperature and humidity (which relates to water in the air). This aligns with general facts.
**Optimal Solution Analysis:** Other options like storms (A) are short-term weather events, mass (C) and density (F) are physical properties, seasonal (D) and winter (E) are time-based categories, length (G) and fluid (H) are unrelated. Climate is the most appropriate and objective answer as it directly encompasses temperature and atmospheric water content over time.

**Fidelity verification:**
"verified": PASS,
"reason":
Step 1: Define the Target: The target is to determine which option (A to H) is correctly described by the terms "temperature and water in the air. "
Step 2: Identify Premises and Conclusion: The premises here are the terms "temperature" and "water in the air". The conclusion is the option that best fits these terms. The argument is that climate is described by these factors.
Step 3: Analyze Argument Structure: The structure is: If a concept is defined by temperature and water in the air, then it's climate.
Step 4: Verify Argument Validity: The premises that climate is described by temperature and water in the air are correct.
Step 5: Consider Counterexamples or Special Cases: No counterexamples refute this relationship.

Figure 7: Case study on the QASC dataset using Qwen3-8B. For this question, the LLM's response is consistent with the ground truth, leading to a PASS in factual verification. In addition, the response also receives a PASS in fidelity verification.

For faithful hallucination detection, the premises here are the terms "temperature" and "water in the air". The conclusion is the option that best fits these terms. If a concept is defined by temperature and water in the air, then it's climate. This is a simple inference. It only needs to combine the two directional facts of "temperature" and "water in the air" to infer the answer climate (option B).

# E  THE USE OF LARGE LANGUAGE MODELS

In the writing stage, we use the large language model to translate and polish. In addition, we use the large language model to detect and modify code errors. We ensure that the design and implementation of the whole idea are not affected by the large language model.

