# OpenReview forum: "F2: Let Large Language Models Think like Aristotle"
_ICLR.cc/2026/Conference — Submitted to ICLR 2026_

### Official Review · Reviewer_gUhG · 2025-10-23

**Soundness:** 2
**Presentation:** 3
**Contribution:** 2
**Rating:** 4
**Confidence:** 3

**Summary:**

This paper proposes F2, a training-free, prompt-based framework designed to detect and correct LLM hallucinations. The key contribution is its novel decomposition of the verification task into two distinct, philosophically-inspired checks: "Factual Hallucination" (verifying correctness and answer optimality) and "Fidelity Hallucination" (verifying the logical structure of the reasoning). The authors present strong empirical results across four datasets and four models, showing significant improvements over baselines like CoT and ToT.

**Strengths:**

1. Originality
The paper's main original contribution is the conceptualization of the F2 framework. Decomposing the self-verification task into two distinct, philosophically-inspired subjects (Factual Hallucination vs. Fidelity Hallucination) is an intuitive and clean heuristic. While the underlying mechanism relies on standard prompt engineering, this specific two-pronged approach to structuring the verification is a creative application of existing ideas.

2. Significance
The work's significance is primarily practical. It offers a useful, training-free, and portable prompting framework that can be immediately applied by developers to improve model reliability.

**Weaknesses:**

1. Limited Technical Novelty

The "F2 mechanics" is not a new algorithm but rather a structured prompting framework for self-verification. It falls within the established paradigm of in-context learning. The novelty lies in the heuristic (Factual vs. Fidelity decomposition), not in the underlying technology.

2. Insufficient and Outdated Baselines

The experimental comparison is limited to foundational (CoT, ToT) and outdated baselines. The paper fails to benchmark F2 against its most direct competitors from 2024-2025, specifically recent works on self-correction, reasoning decomposition, and verifier-based methods [1, 2, 3]. This makes it impossible to assess its true contribution.

3. Critical Gaps in Experimental Protocol (Unfair Comparison)

This is the most significant flaw. The F2 method's "Retry Mechanism" (Section 3.4) constitutes a form of rejection sampling, effectively giving it a $pass@k$ advantage (where $k \ge 2$). If you set a lower threshold for baseline methods, it is basically making fewer retries for baselines. However, I cannot find details on how you set the acceptance threshold of the baseline methods.

Without this clarification, the reported performance gains are confounded by a larger, unacknowledged sampling budget, and the central claim of the paper's superiority is unsubstantiated.

References:

[1] Zhenyu Wu, Qingkai Zeng, Zhihan Zhang, Zhaoxuan Tan, Chao Shen, and Meng Jiang. 2024. Large Language Models Can Self-Correct with Key Condition Verification. In Proceedings of EMNLP 2024.

[2] Junyu Luo, Cao Xiao, and Fenglong Ma. 2024. Zero-Resource Hallucination Prevention for Large Language Models. In Findings of ACL: EMNLP 2024.

[3] Xiaoying Zhang, Baolin Peng, Ye Tian, Jingyan Zhou, Lifeng Jin, Linfeng Song, Haitao Mi, and Helen Meng. 2024. Self-Alignment for Factuality: Mitigating Hallucinations in LLMs via Self-Evaluation. In Proceedings of ACL 2024.

**Questions:**

Thank you for your work. I find the F2 framework to be an intuitive and clearly-presented approach. However, I have several critical questions, primarily regarding the experimental methodology, that would need to be addressed to validate the paper's claims. Answering these could significantly clarify my concerns.

1. On the "Retry Mechanism" and Fair Experimental Comparison:

This is my most significant concern. The F2 method's "Retry Mechanism" (Section 3.4) constitutes a form of rejection sampling. This appears to give F2 a $pass@k$ advantage (where $k \ge 2$), while the baselines (Zero-shot, CoT, ToT) are not known. The reported performance gains may stem from this larger, unacknowledged sampling budget rather than the F2 verification framework itself.

Q1.1 (Fairness): Is my understanding correct? Were the Zero-shot, CoT, and ToT baselines evaluated at different tolerance levels? How are those methods applied, and how is the threshold set?

Q1.2 (CoT-SC Details): For the CoT-SC baseline, how many thought chains (samples) were used for its majority vote? How does this total sampling budget compare to the average number of samples (initial sample + retries) used by F2?

2. On Insufficient and Outdated Baselines:

The experimental comparison is limited to foundational (CoT, ToT) and outdated baselines. The paper fails to benchmark F2 against its most direct competitors from 2024-2025.

Q2.1: Could you please discuss how your F2 framework compares to recent methods on self-correction, reasoning decomposition, and verifier-based methods, such as Wu et al. (2024) [1], Luo et al. (2024) [2], and Zhang et al. (2024) [3]? This context is essential to assess your work's contribution.

3. On the Nature of the Contribution (Technical Novelty):

The paper frames F2 as a new "mechanics." However, the implementation (Figs 5 & 6) appears to be a sophisticated, multi-step prompting framework that falls within the established paradigm of in-context learning.

Q3.1: Could you clarify what is mechanically novel about the method beyond this specific prompt structure? Is the contribution a new algorithm or a new (and effective) heuristic for in-context learning?

References:

[1] Zhenyu Wu, Qingkai Zeng, Zhihan Zhang, Zhaoxuan Tan, Chao Shen, and Meng Jiang. 2024. Large Language Models Can Self-Correct with Key Condition Verification. In Proceedings of EMNLP 2024.

[2] Junyu Luo, Cao Xiao, and Fenglong Ma. 2024. Zero-Resource Hallucination Prevention for Large Language Models. In Findings of ACL: EMNLP 2024.

[3] Xiaoying Zhang, Baolin Peng, Ye Tian, Jingyan Zhou, Lifeng Jin, Linfeng Song, Haitao Mi, and Helen Meng. 2024. Self-Alignment for Factuality: Mitigating Hallucinations in LLMs via Self-Evaluation. In Proceedings of ACL 2024.

---

### Official Review · Reviewer_KCGZ · 2025-10-25

**Soundness:** 1
**Presentation:** 2
**Contribution:** 1
**Rating:** 2
**Confidence:** 4

**Summary:**

The authors propose F2, a novel framework for hallucination detection in LLMs inspired by Aristotelian philosophy. F2 decomposes hallucination into two complementary dimensions. The first, Factual Hallucination Detection, focuses on verifying the factual correctness of generated content and identifying the most appropriate answer. The second, Fidelity Hallucination Detection, examines the logical validity of reasoning by applying classical logical structures such as syllogistic reasoning.
The framework operates as a training-free, self-verifying approach, making it easily applicable to a wide range of existing LLMs. The authors evaluate F2 on multiple benchmarks, including CommonsenseQA, QASC, StrategyQA, and HaluEval, and demonstrate consistent improvements over baseline methods such as CoT, CoT-SC, ToT, and zero-shot inference.

**Strengths:**

1. The paper introduces a conceptually engaging two-part decomposition of hallucination, linking it explicitly to classical Aristotelian reasoning. This separation between factual and fidelity hallucination detection provides a clear analytical lens for diagnosing the underlying causes of model errors.

2. Across four benchmark datasets and multiple LLM architectures, the empirical results demonstrate consistent performance gains. F2 substantially outperforms zero-shot and advanced prompting baselines such as CoT, CoT-SC, and ToT, underscoring its robustness and general applicability.

3. Moreover, the framework is model-agnostic and training-free, allowing it to be easily integrated into diverse LLM applications without fine-tuning or architectural modification.

**Weaknesses:**

1. The paper does not include comparative experiments with recent hallucination detection methods and fails to establish a clear distinction from existing detection frameworks. While reasoning-related baselines (e.g., CoT, ToT) are included, recent hallucination detection methods are not.

2. The proposed Methodology (Section 3) lacks scientific grounding and algorithmic specificity. Most equations are presented as symbolic definitions without concrete implementation details. The “retry mechanism” in Section 3.4 is especially vague — it is unclear what qualifies as a “concise reason,” how many retry iterations are performed, or what stopping criteria are applied. Greater algorithmic transparency is necessary to ensure reproducibility and to elevate the framework beyond a descriptive or prompt-engineering level.

3. The mathematical modeling of F2 is not formulated with sufficient rigor.
- Equation (1) does not specify input–output domains, making its functional definition ambiguous.
- Equation (2) lacks a clear definition of “optimal solution”.
- Equations (3)–(7) merely represent logical structures without any computational process
In practice, they correspond to prompt templates imitating logical reasoning, not formal inference mechanisms.

4. The evaluation protocol is not clearly reproducible. Key details such as prompt formatting, extraction rules, and answer parsing are confined to the appendix rather than presented in the main text, leaving doubts about the replicability of results.

5. The framework has not been validated on open-ended generation tasks, where hallucination tends to manifest more subtly. This limitation restricts the claimed generality and cross-domain applicability of F2.

6. Given the dual-verification and retry structure, significant computational overhead is likely, yet the authors provide no analysis of time or cost. Reporting such resource implications is important for assessing the practical feasibility of the method.

7. The paper’s writing flow is often uneven, with awkward phrasing and inconsistent transitions, which somewhat diminishes readability and overall polish.

8. Finally, there is a mismatch between the logical structure presented and its actual implementation. Figure 2 visually suggests a formal logic verification pipeline, but in reality, the process appears to rely solely on prompt-based reasoning by the LLM, without any symbolic or automated logical validation.

**Questions:**

1. Line 19, we proposes --> we propose
2. Line 276, Our experiments was  --> were
3. Line 477, a new hallucination detection mechanics --> mechanism

---

### Official Review · Reviewer_kXKt · 2025-11-01

**Soundness:** 2
**Presentation:** 3
**Contribution:** 2
**Rating:** 2
**Confidence:** 4

**Summary:**

This paper proposes a new prompting-based hallucination detection method for Large Language Models (LLMs). Instead of relying on external knowledge bases or large annotated datasets, the method guides LLMs to reason like Aristotle by decomposing hallucination detection into two parts: (1) factual hallucination detection, which verifies the truthfulness of generated content, and (2) fidelity hallucination detection, which applies classical logical reasoning such as syllogism to assess logical consistency. Experiments show that this approach enhances LLMs’ ability to identify hallucinations, improves interpretability, and helps developers locate and correct model errors.

**Strengths:**

1. The method introduces an original, interpretable approach by modeling LLM reasoning after Aristotle’s logic, bridging philosophy and modern AI.

2.  Its self-reflection feature avoids reliance on external knowledge bases or large annotated datasets, improving efficiency and generalization across domains.

3.The two-stage (factual and logical) decomposition provides clearer insights into the sources of hallucinations, aiding model debugging and improvement.

**Weaknesses:**

1. In Table 1, the results are grouped by prompting methods rather than by models, which makes it difficult to compare model performance effectively. Reorganizing the table by models would improve readability and clarity.

2. The ablation study did not include HaluEval.

3. In Section 3.4, the paper mentions using F2 as a verifier with a retry mechanism to improve performance, but it is unclear how many retries are allowed or whether there are constraints on computation time or resources.

4. The evaluation could be improved by including comparisons with commonly studied self-reflection or self-verification methods such as [1,2], to better position the proposed approach within existing literature.

[1]https://arxiv.org/pdf/2303.11366
[2]https://arxiv.org/pdf/2210.03629

**Questions:**

Related Works:
1. https://aclanthology.org/2024.naacl-long.424.pdf


Typos:

1. Line 306: There should be a space between ``methods:'' and  ``zero-shot''

---

### Official Review · Reviewer_xs1d · 2025-11-01

**Soundness:** 2
**Presentation:** 3
**Contribution:** 2
**Rating:** 4
**Confidence:** 2

**Summary:**

This paper introduces a novel hallucination detection framework inspired by the cognitive processes of Aristotle, which decomposes hallucination detection into two key components: factual hallucination detection and fidelity hallucination detection. The method aims to improve the reliability of Large Language Models (LLMs) by enabling them to evaluate the truthfulness and logical consistency of their outputs. The framework uses a self-verification approach without requiring external resources or extensive training, and it demonstrates significant improvements in performance on various benchmark datasets. Experimental results show that the method outperforms existing hallucination detection techniques, enhancing both recognition and reasoning capabilities of LLMs.

**Strengths:**

1) The novel approach of combining factual and fidelity hallucination detection offers a comprehensive method for addressing both the truthfulness and logical consistency of LLM-generated content.

2) The self-verification framework is resource-efficient, requiring no additional training or external knowledge bases, making it a lightweight solution for hallucination detection.

3) Extensive experimental results show significant improvements in accuracy on multiple benchmark datasets, demonstrating the method's effectiveness and robustness.

**Weaknesses:**

1) While the method performs well on the tested datasets, its generalizability to other types of hallucinations or more complex reasoning tasks remains unclear and requires further exploration.

2) The reliance on internal model mechanisms for hallucination detection may lead to limitations in identifying subtle or more complex hallucinations that are not immediately evident in model outputs.

**Questions:**

See weaknesses.

---

### Meta-Review · Area_Chair_px7B · 2026-01-01

**Summary:**

Reviewers agree that the proposed F2 framework lacks sufficient technical novelty beyond standard prompting and relies on an unfair evaluation protocol—specifically the retry mechanism—which precludes a valid comparison with relevant sota baselines.

**Reviewer Concerns:**

While the authors seemingly addressed minor issues regarding table organization and typos, the critical concerns regarding the "retry mechanism" functioning as an unacknowledged rejection sampling advantage and the absence of comparisons to recent 2024–2025 self-correction methods remain outstanding.

**Reviewer Scores:**

Reviewers would likely align with the rejection consensus had they fully engaged with the critical flaws in the experimental design highlighted by the other reviewers.

---

### Decision · Program_Chairs · 2026-01-26

Reject